# H₂O₂ gel bleaching induces cytotoxicity and pain conduction in dental pulp stem cells via intracellular reactive oxygen species on enamel/dentin disc

**Chang Chen[1☯], Xiansheng Huang[2☯], Wenqiang Zhu[2], Chen Ding[2], Piaopiao Huang[2], Rong Li[1]***

**1** Department of Stomatology, The Second Xiangya Hospital, Central South University, Changsha, Hunan, China, **2** Department of Cardiovascular Medicine, The Second Xiangya Hospital, Central South University, Changsha, Hunan, China

☯ These authors contributed equally to this work.
* rongli@csu.edu.cn

**Data Availability Statement:** All relevant data are within the paper and its Supporting information files.

## Abstract

### Background

Bleaching is widely accepted for improving the appearance of discolored teeth; however, patient compliance is affected by bleaching-related complications, especially bleaching sensitivity. This study aimed to investigate the role of reactive oxygen species (ROS) in cytotoxicity and pain conduction activated by experimental tooth bleaching.

### Methods

Dental pulp stem cells with or without N-acetyl-L-cysteine (NAC), an ROS scavenger, were cultured on the dentin side of the enamel/dentin disc. Subsequently, 15% (90 min) and 40% (30 min) bleaching gels were painted on the enamel surface. Cell viability, intracellular ROS, $Ca^{2+}$, adenosine triphosphate (ATP), and extracellular ATP levels were evaluated using the Cell Counting Kit-8 assay, 2',7'-dichlorodihydrofluorescein diacetate, CellROX, fura-3AM fluorescence assay, and ATP measurement kit. The rat incisor model was used to evaluate in vivo effects after 0, 1, 3, 7, and 30 days of bleaching. Changes in gene and protein expression of interleukin 6 (IL-6), tumor necrosis factor-alpha (TNFα), transient receptor potential ankyrin 1 (TRPA1), and Pannexin1 (PANX1) in dental pulp stem cells and pulp tissue were detected through RT-PCR, western blotting, and immunofluorescence.

### Results

The bleaching gel suppressed dental pulp stem cell viability and extracellular ATP levels and increased intracellular ROS, $Ca^{2+}$, and intracellular ATP levels. The mRNA and protein expression of IL-6, TNFα, TRPA1, and PANX1 were up-regulated in vitro and in vivo. Furthermore, the 40% gel had a stronger effect than the 15% gel, and NAC ameliorated the gel effects.

**Funding:** This work was financially supported by National Nature Science Foundation of China (Nos. 81700999 and 81974281) and Science Foundation of Hunan Province (Nos. 2018JJ3741 and 2020JJ2052).

**Competing interests:** The authors declare that they have no competing interests.

## Conclusions

Our findings suggest that bleaching gels induce cytotoxicity and pain conduction in dental pulp stem cells via intracellular ROS, which may provide a potential therapeutic target for alleviating tooth bleaching nociception.

## Introduction

Given the increasing popularity of esthetic procedures, tooth bleaching is among the most widely performed treatments to achieve a pleasing appearance [1]. Although tooth bleaching allows an aesthetically pleasing smile, bleaching-induced tooth sensitivity (bleaching sensitivity [BS]) remains its most common clinical side effect, affecting >70% of the patients [2, 3]. BS can occasionally cause painful or uncomfortable sensations, which affects the patient's satisfaction with the bleaching experience; moreover, severe BS can lead to treatment discontinuation [4].

The mechanisms underlying BS remain unclear. It has been suggested that BS may be associated with bleaching products [5], which are generally based on hydrogen peroxide ($H_2O_2$) or carbamide peroxide with concentrations between 3% and 40% [6]. $H_2O_2$ can penetrate the enamel and dentin tubules to reach the pulp, leading to reactive oxygen species (ROS) production, which decreases cellularity and cellular metabolism, alters vascular permeability, and even causes tissue necrosis [7, 8]. Hence, activation of dental pulp inflammation and oxidative stress may be involved in tingling/shooting pain without provoking stimuli during or after tooth bleaching.

Many recent studies have shown that $H_2O_2$-induced pain is associated with the transient receptor potential ankyrin 1 (TRPA1) [9, 10]. TRPA1 is a nonselective cation channel that is primarily expressed in nociceptive neurons and can be activated by noxious cold, ROS, and mechanical stimuli [11–13]. Increasing evidence has shown that TRPA1 is activated by various oxidizing compounds, including $H_2O_2$, and causes pain [10, 14]. Moreover, it was recently reported that TRPA1 activates nociceptors in dissociated vagal neurons and peripheral terminals of bronchopulmonary vagal afferents, which can be mediated by ROS [14, 15]. These novel findings are indicative of the role of ROS and TRPA1 in nociceptor activation under high $H_2O_2$ concentrations.

$Ca^{2+}$ influx mediated by the $Ca^{2+}$ channel (including TRPA1) is critical in nerve excitement generation and conduction. It not only causes the fusion of synaptic vesicles with the presynaptic membrane to release neurotransmitters but also acts as a second messenger for regulating the activity of intracellular enzymes and non-enzymatic proteins. Further, it is involved in regulating cell signal transduction by altering intracellular ion levels [16]. Neural pain occurrence and conduction are closely associated with the gap junction hemichannel protein, Pannexin1 (PANX1); moreover, $Ca^{2+}$ influx leads to PANX1 activation [17]. Consequently, it promotes the production and release of adenosine triphosphate (ATP) [18], an important neurotransmitter that activates the purinergic receptor P2X of central or peripheral nerve cells, induces nerve cell membrane depolarization, excites sensory neurons, and is involved in pain generation and conduction [19].

However, the mechanism underlying TRPA1 in BS remains unclear. TRPA1 was recently detected in freshly isolated human dental pulp [20, 21], and its expression was increased in pulpal inflammation or injury [20]. We speculated that ROS excite or sensitize pulpal nociceptors, including TRPA1, and cause BS. However, there has been no study on the relationship between the expression and functional significance of TRPA1 and ROS in the dental pulp

during bleaching treatment. This study aimed to investigate the cytotoxicity of ROS generated by different H$_2$O$_2$ concentrations on dental pulp stem cells (DPSCs), as well as the post-bleaching changes in pain conduction.

## Materials and methods

### Cell culture

Cultured human DPSCs were acquired from premolars freshly extracted for orthodontic treatment (aged 18–24 years), following the protocol approved by the Ethics Committee of the Second Xiangya Hospital, Central South University, China. After tooth extraction, the pulp was aseptically and immediately obtained, cut into small pieces (about 1 mm$^3$), and incubated for 24 h in complete Dulbecco's Modified Eagle Medium (DMEM; supplemented with 100 IU/mL penicillin, 100 mg/mL streptomycin, 2 mmol/L glutamine; Gibco, NY, USA), which contained 10% fetal bovine serum (FBS; Gibco, NY, USA) and 200 units/mL of type II collagenase (Worthington Biochemical Corporation, NJ, USA). Next, the liberated cells were cultured in complete DMEM supplemented with 10% FBS. Cells at the third passage were used in the follow-up experiment as previously described [22], with a slight modification being that the cells were implanted on the dentin surface of an enamel/dentin disc (Fig 1A).

### Enamel/dentin specimen preparation

Intact premolars were acquired from individuals aged 18–24 years after obtaining consent. Before extraction, the tooth surfaces were cleaned with pumice stone and water using a low-speed handpiece. First, dental crowns were obtained by horizontally sectioning the tooth under the cementoenamel junction using a low-speed diamond saw (Isomet, Buehler, IL, USA). Next, the lingual crowns were removed with a specimen containing an integrated disc of buccal enamel-dentin being maintained. Subsequently, the lingual disc layer was heaped using light-cured resin (Filtek P60, 3M ESPE) to form a bottomless cylinder (Fig 1A). Before the *in vitro* experiments, all specimens were sterilized using 75% ethanol and ultraviolet light irradiation for 1 h [8].

### Experimental design

DPSCs were seeded onto the dentin surface of the disc and added to 24-well plates for 24 h in DMEM supplemented with 10% FBS. Subsequently, the culture medium was replaced with DMEM without FBS. Next, the enamel/dentin specimen was inverted in wells with the enamel surface exposed above the liquid level; further, the whole enamel surface was exposed to the bleaching gel at 37°C. This study tested two H$_2$O$_2$ concentrations, including a 40% and 15% H$_2$O$_2$ gel (Opalescence PF; Ultradent Products, UT, USA) [23]. Approximately 0.5 mm thickened gel was spread on the enamel surface [8]. The following groups were used: G1—untreated DPSCs, the negative control group; G2—DPSCs treated with 1 mM N-acetyl-L-cysteine (NAC) for 2 h; G3—DPSCs treated for 90 min with 15% H$_2$O$_2$ gel; G4—DPSCs treated for 90 min with 15% H$_2$O$_2$ gel after pretreatment with 1 mM NAC for 2 h; G5—DPSCs treated for 30 min with 40% H$_2$O$_2$ gel; and G6—DPSCs treated for 30 min with 40% H$_2$O$_2$ gel after pretreatment with 1 mM NAC for 2 h [24, 25]. Immediately after the bleaching process ended, the disc was re-inverted, followed by cell analysis.

### Cell identification

DPSC characterization was performed through flow cytometry using specific anti-human antibodies against CD14, CD20, CD34, CD45, CD73, CD90, and CD105 (eBioscience, CA, USA)

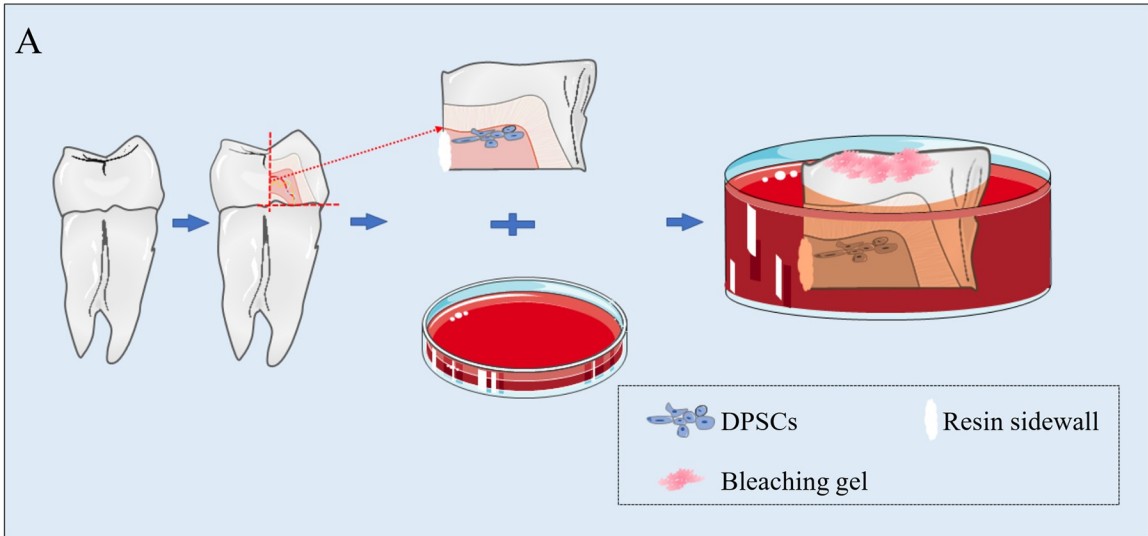

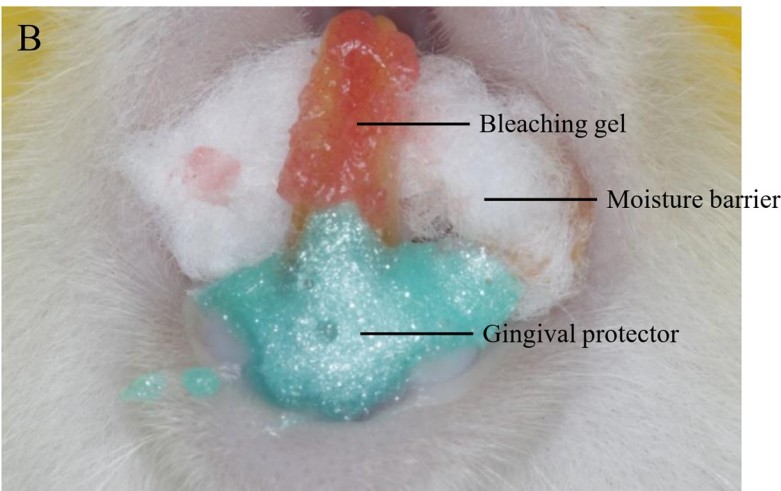

**Fig 1. Schematic diagram (A) of the enamel-dentin disc and rat incisor model (B).**

as recommended by the International Society for Cellular Therapy. Additionally, we measured the DPSC multi-differentiation potency toward osteogenesis and adipogenesis. Osteogenesis was induced after culturing cells in osteogenic media (Cyagen, CA, USA) for 25 days. The cells were fixed in 4% paraformaldehyde (PFA) and calcium-rich deposits with Alizarin Red solution (Sigma-Aldrich, MO, USA). To allow adipogenesis, DPSCs were cultured in adipogenic media (Cyagen, CA, USA) for 14 days. Subsequently, DPSCs were fixed with 4% PFA and neutral lipid-containing cytoplasmic vacuoles with Oil Red O solution (Sigma-Aldrich, MO, USA).

## Cell cytotoxicity analysis

Cell cytotoxicity was measured using the Cell Counting Kit-8 (CCK8) (Dojindo, Japan) assay following the manufacturer's instructions. After bleaching, the DPSCs were implanted in 96-well plates ($6 \times 10^3$ cells/well) overnight for adherence. After 24 h, cells were gently cleaned using phosphate-buffered saline (PBS), and the medium was replaced using fresh DMEM

supplemented with 10% CCK-8. Next, cells were incubated at 37˚C for 2 h. Absorbance at 450 nm was measured using a spectrophotometer (HEALES, Shenzhen, China) [26].

### ROS assay

Oxidative stress was immediately tested after bleaching, using a 5 mM cell-permeant fluorescence probe, 2',7'-dichlorodihydrofluorescein diacetate (H₂DCFDA) (Life Technologies, CA, USA) or 5 µM CellROX (Thermo, MA, USA) at 37˚C for 30 min. The culture medium was aspirated and washed using PBS, followed by detection using flow cytometry. The mean fluorescence intensity (MFI) was used for statistical analysis [27].

### Real-time PCR

We measured mRNA expression of tumor necrosis factor-alpha (TNF-α), interleukin 6 (IL-6), TRPA1, and PANX1 using real-time PCR as previously described [8, 22]. Total RNA was extracted with Trizol (Thermo, MA, USA), followed by reverse transcription to single-stranded cDNA using a HiFiScript cDNA Synthesis Kit (CWBIO, Beijing, USA), according to the manufacturer's protocols. We used TaqMan assays (CWBIO, Beijing, USA) for human genes. Relative expression levels were normalized using the $2^{-\Delta\Delta ct}$ method and housekeeping gene actin. Table 1 shows the primer sequences.

### Western blotting

Western blotting was performed as previously described [28]. Equal protein amounts were separated through sodium dodecyl sulfate-polyacrylamide gel electrophoresis using a Bio-Rad system and 15% acrylamide gel after dilution with sample buffer. Next, the proteins were transferred onto nitrocellulose membranes using a Bio-Rad system. Subsequently, the membrane was blocked using 10% non-fat dry milk in 25 mM Tris-buffered saline (pH 7.2) and 0.1% Tween 20 (TBST) for 2 h, and then incubated with the primary antibody to TRPA1 (1:1000, Novus Biologicals, CO, USA), TNFα (1:1000, Abcam, Cambridge, UK), IL6 (1:1000, Abcam, Cambridge, UK), PANX1 (1:2000, Proteintech, IL, USA), and β-actin (1:5000, Proteintech, IL, USA). After washing in TBST, the membrane was incubated using horseradish peroxidase-conjugated goat anti-rabbit IgG (1:5000, Proteintech, IL, USA) for 1 h at 37˚C. After a final wash, we used enhanced chemiluminescence (ECL) for signal detection using a WesternBright ECL kit (Advansta, CA, USA).

### Intracellular Ca²⁺ measurements

Ca²⁺ was fluorometrically loaded using the Ca²⁺ probe fura-3AM (Beyotime, Shanghai, China). After bleaching, the cells were stained with fura-3AM (5 M) for 20 min at 37˚C and analyzed using flow cytometry. MFI was used for statistical analysis.

**Table 1. Primer sequences.**

| Gene | Forward (5' to 3') | Reverse (5' to 3') |
|------|--------------------|--------------------|
| actin | ACCCTGAAGTACCCCATCGAG | AGCACAGCCTGGATAGCAAC |
| TNFα | GAACCCCGAGTGACAAGCCT | TATCTCTCAGCTCCACGCCAT |
| IL6 | GCAATAACCACCCCTGACCCAA | GCTACATTTGCCGAAGAGCC |
| TRPA1 | GCATGTTTATTCCCTCACTACCCC | ACACAAGGACACATACATAGCCA |
| PANX1 | ACTTGGTTTCCCCGCATGGT | GAACAAAGCGCTTCCCTCTGG |

## ATP release

Intracellular and extracellular ATP levels were measured using an ATP Content Assay kit (Jiancheng, Nanjing, China) as per the manufacturer's protocol. The ATP levels were calculated from the ATP standard curve [19].

## Animal experiments

For in vivo experiments, we used 36 male and female 8 weeks old Sprague Dawley rats with a mean weight of 250–350 g after approval by the Ethics Committee of the Second Xiangya Hospital, Central South University, China, with three rats as an experimental unit. The rats were cultivated in a standard animal room at 22 ± 1˚C, 55 ± 10% humid atmospheres, with a standard light/dark schedule of food and water. Three groups were used: 1) no bleaching gel as control (one experimental unit), 2) 15% bleaching gel for 90 min (five experimental units), and 3) 40% bleaching gel for 30 min (five experimental units). Grouping was done by simple randomization; that is, the samples were numbered sequentially and then grouped. The different groups were kept in separate cages to minimize potential confounders. Bleaching gel (0.01 mL) was smeared onto the incisor surface following the manufacturer's instructions (Fig 1B). At 0, 1, 3, 7, and 30 days of bleaching, experimental rats (n = 3) and a control group were anesthetized with pentobarbital (80 mg/kg) to observe the effect of bleaching on the pulp tissue at different times; subsequently, they were sacrificed. The incisor was fixed in 4% neutral buffered formalin for 24 h; subsequently, it was demineralized in 10% ethylenediaminetetraacetic acid solution for 3 months. Next, the samples were embedded in paraffin. Three-micrometer sections were obtained from the specimens, followed by analysis using standard hematoxylin and eosin staining for routine histopathological assay. Nine rats were used in each experiment, a total of 33 rats were used in the experiment, and three rats were spared. No rats died, and there were no exclusions. This experiment used a three-blind method; that is, bleaching, sampling and slicing, and data analyses were completed by different personnel.

## Immunofluorescence

Dewaxed slices were treated with 0.1% H₂O₂ and 1 mg/mL trypsin for 15 min and 30 min, respectively. After incubation with 10% goat serum for 30 min at room temperature, the slices were treated using rabbit polyclonal antibody against TNF-α (Bioss, Beijing, China), IL6 (Proteintech, IL, USA), TRPA1 (Novus Biologicals, CO, USA), and PANX1 (Proteintech, IL, USA) overnight at 4˚C. For immunohistochemistry, the slices were treated using CoraLite488–conjugated Affinipure Goat Anti-Rabbit IgG (Proteintech, IL, USA) and cleaned with PBS. For nuclear staining, we used DAPI at room temperature for 5 min.

## Statistical analysis

All experiments were repeated at least three times. Data are expressed as the mean ± SD. Statistical analyses were performed using the GraphPad Prism 6.0 (GraphPad Software, CA, USA) and unpaired Student's $t$-tests. Graphs were drawn using Graphpad Prism 6.0. All levels of statistical significance were set at $P < 0.05$.

## Results

### DPSC characterization

Flow cytometry showed that DPSCs were positive for CD73, CD90, and CD105 and negative for CD14, CD20, CD34, and CD45 (Fig 2A). Moreover, DPSCs showed apparent alizarin red-stained mineralized nodules (Fig 2B) and oil red O-stained lipid clusters (Fig 2C).

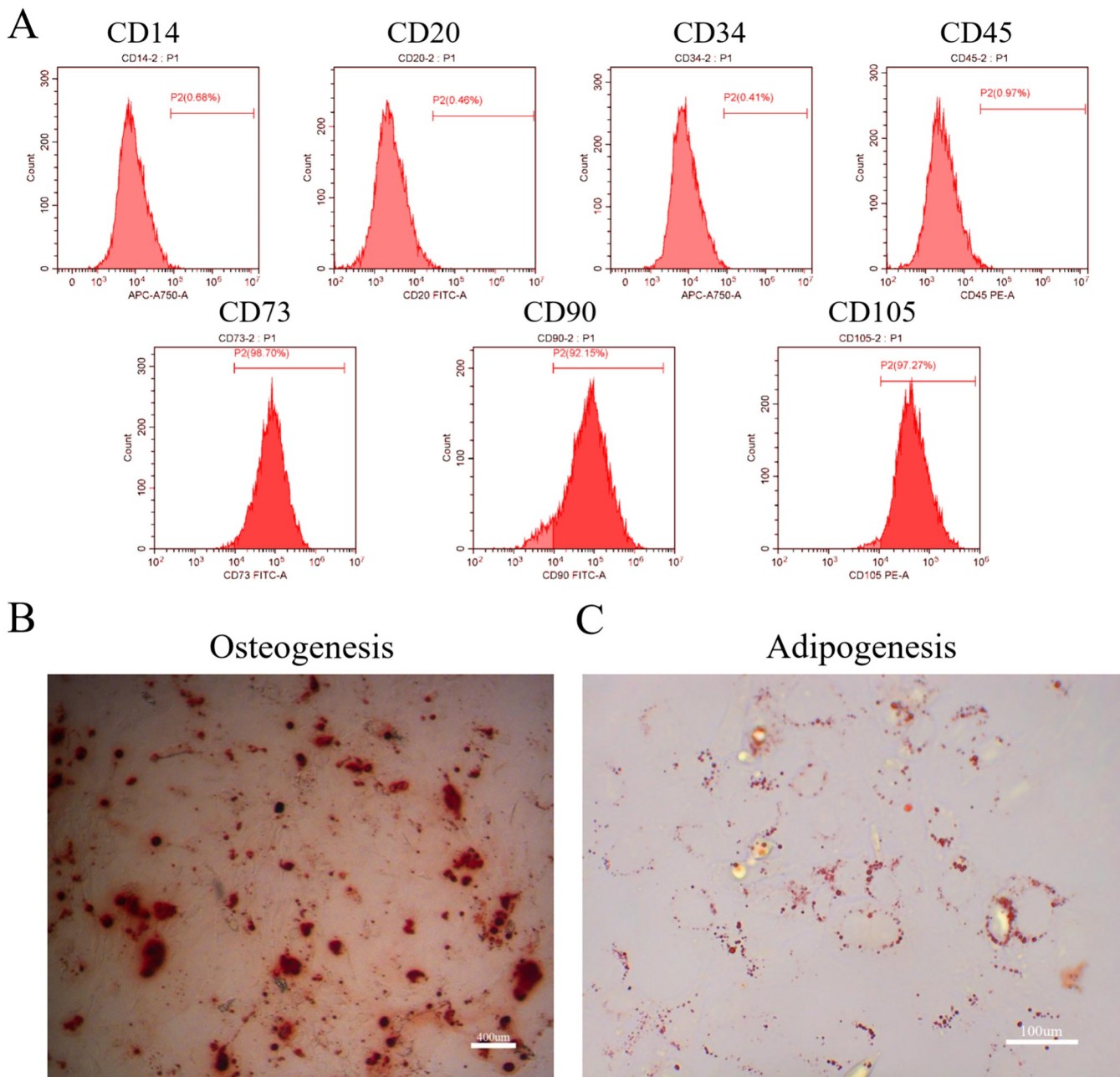

**Fig 2. Identification of dental pulp stem cells (DPSCs).** Flow-cytometry (A) illustrates the immunophenotypic characteristics of human DPSCs. Human DPSCs were positive for cell surface antigens CD73, CD90, and CD105, as well as negative for CD14, CD20, CD34, and CD45. DPSCs were cultured under osteogenic (B, 14 days) or adipogenic (C, 21 days) conditions and showed mineralized nodules and lipid clusters as revealed by alizarin red and oil red staining, respectively. Scale bar = 400 (B) or 100 (C) μm.

### Forty percent bleaching gel displayed higher ROS-mediated cytotoxicity

The CCK-8 assay was used to test the cytotoxicity at different bleaching gel concentrations and the role of ROS (Fig 3A and 3B). Bleaching gel significantly suppressed DPSC viability; further, compared with 15% gel, 40% gel exerted a stronger inhibiting effect. The addition of NAC, which is an ROS scavenger, partly improved the cell viability proportionally to the inhibition

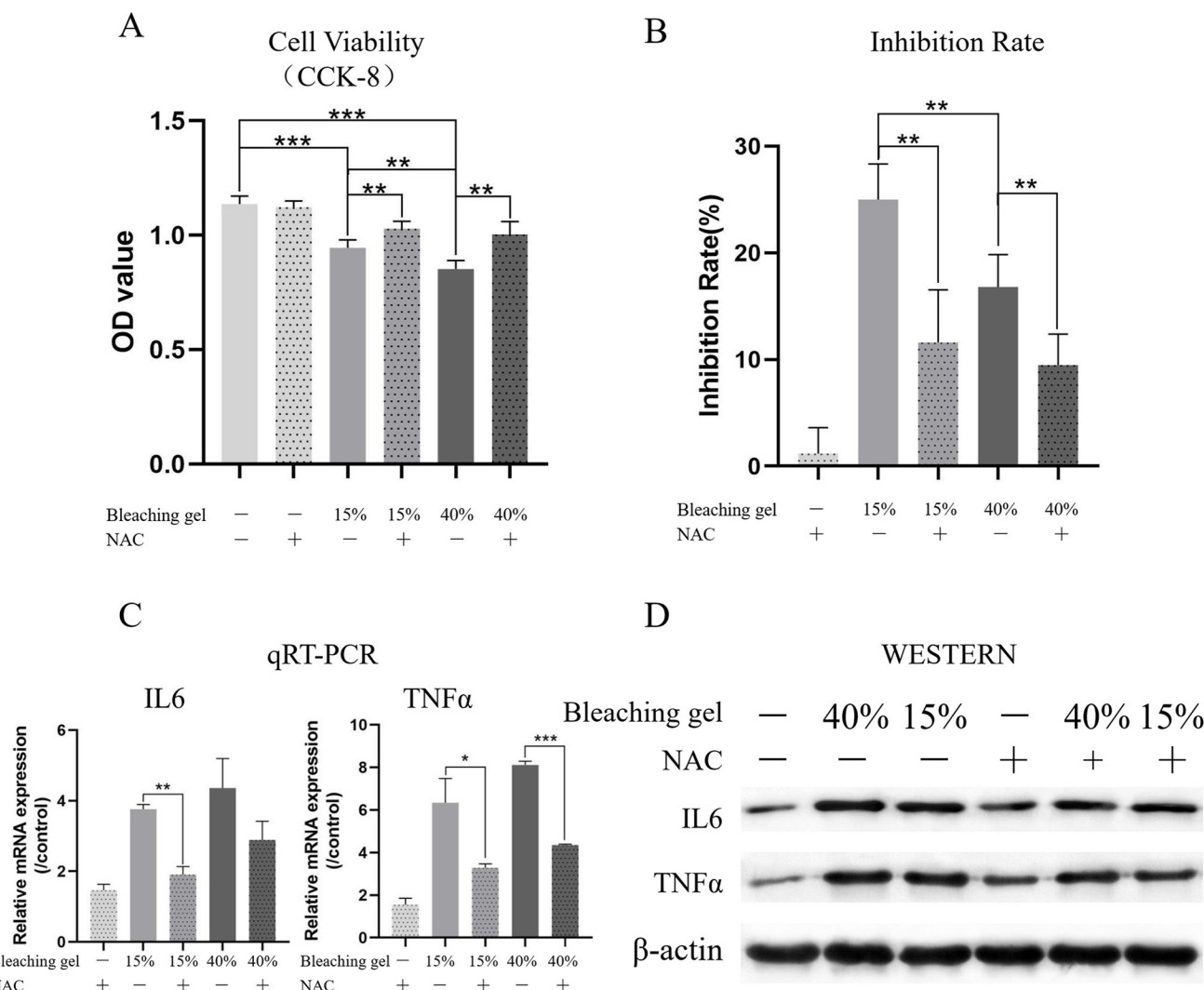

**Fig 3. 40% bleaching gel displays higher ROS-mediated cytotoxicity.** After bleaching using 40% and 15% H$_2$O$_2$ gel for 30 min and 90 min, respectively, through the enamel/dentin disc, cell viability (A) and inhibition rate (B) were detected using the CCK-8 assay. Compared with the 15% gel, the 40% gel showed stronger viability inhibition, which was reverted partly by adding NAC (an ROS scavenger). The between-group differences in mRNA (C) and protein (D) expression of inflammatory factors IL-6 and TNFα were consistent with the viability findings. $^{**}$p<0.01, $^{***}$p<0.001. IL-6, interleukin 6; ROS, reactive oxygen species; TNFα, tumor necrosis factor-alpha.

degree. Subsequently, we performed quantitative reverse transcription-polymerase chain reaction (qRT-PCR) (Fig 3C) and western blotting (Fig 3D) to determine the expression levels of IL-6 and TNFα as representative inflammatory cytokines, which revealed an analogous between-group variance.

## ROS mediate pain conduction in DPSCs caused by the bleaching gel

To probe the pain conduction effect of bleaching gel, we examined intracellular levels of ROS, Ca$^{2+}$, and ATP; further, we used the MFI to calculate the diversity. We found that bleaching gel led to a dose-dependent increase in ROS (Fig 4A–4D) and Ca$^{2+}$ levels (Fig 4E and 4F) in DPSCs. Moreover, this increase was attenuated by NAC. Similar changes occurred in the Ca$^{2+}$ inflow channel TRPA1 and ATP excretion channel PANX1 (Fig 4I and 4J). Notably, the

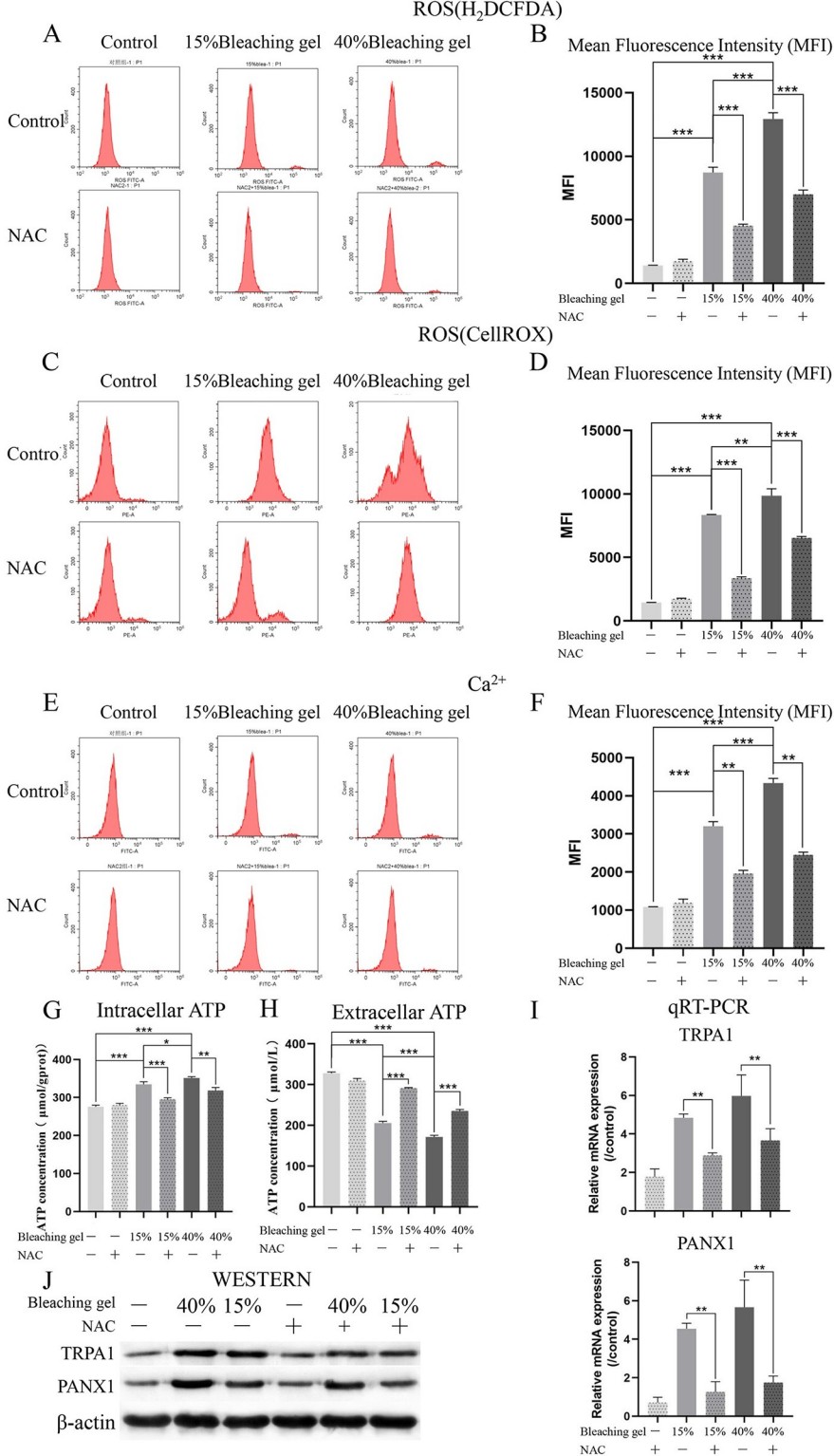

**Fig 4. ROS mediate pain conduction in DPSCs caused by the bleaching gel.** Cells underwent different treatments, and intracellular ROS (A, B by H₂DCFDA and C, D by CellROX) and Ca²⁺ (E, F) were measured using flow cytometry. Mean fluorescence intensity was used to quantify between-group differences. The bleaching gel increased intracellular ROS and Ca²⁺, with the 40% gel exerting a more intense effect; NAC weakened this trend. Intracellular and extracellular ATP (G, H) levels were detected using the ATP Assay Kit, which revealed that they completely differed.

The pain conduction critical tunnel TRPA1 and PANX1 expression levels of mRNA and protein measured by qRT-PCR (I) and western blotting (J) displayed a homologous drift. $^{**}p < 0.01$, $^{***}p < 0.001$. ATP, adenosine triphosphate; DPSC, dental pulp stem cell; NAC, N-acetyl-L-cysteine; PANX1, Pannexin1; ROS, reactive oxygen species; qRT-PCR, quantitative reverse transcription- polymerase chain reaction; TRPA1, transient receptor potential ankyrin 1.

extracellular ATP levels (Fig 4G) were different from the intracellular ATP levels (Fig 4H), which could be attributed to H₂O₂ hydrolysis in the gel on ATP and could have affected the accuracy of the results.

## Histological analysis

Bleaching gel was placed on the labial surfaces of rat incisors for 30 min (40% gel) and 90 min (15% gel); subsequently, the teeth were visualized using HE staining (Fig 5) after 0, 1, 4, 7, and 30 days. The control group showed an integrated pulp surrounded by continuous odontoblastic layers and predentin. Contrastingly, the bleaching groups showed more and expanded vessels. Furthermore, specimens in the 40% gel group revealed some interspace, which was considered as necrosis.

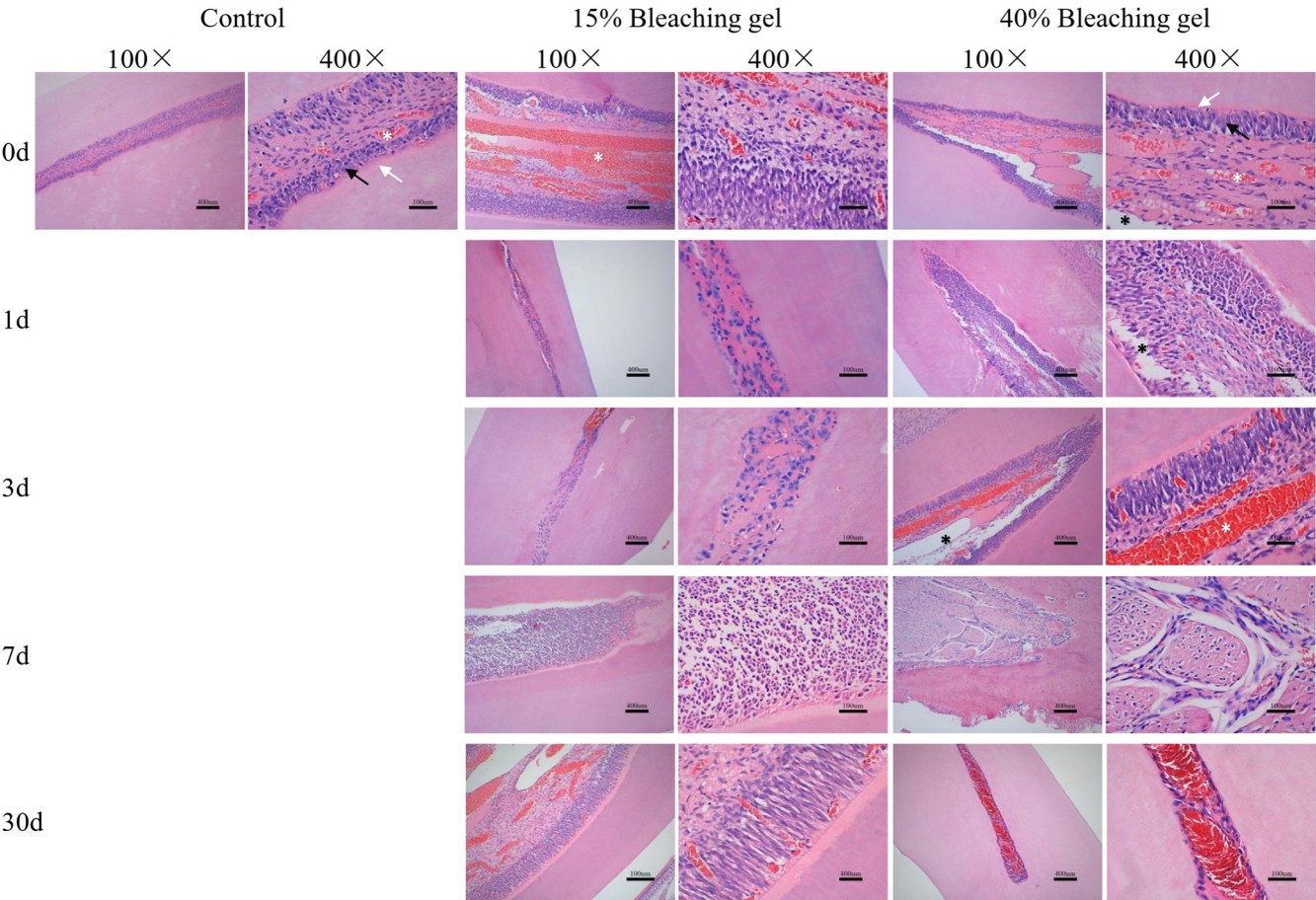

**Fig 5. HE staining of rat incisors after gel bleaching.** Rat incisors were bleached with the 15% or 40% gel for 90 or 30 min, respectively. After 0, 1, 3, 7, and 30 days, HE staining was used to analyze histological changes. White arrowhead, predentin layer; black arrowhead, odontoblastic layer; white asterisks, vessels; black asterisks, necrotic areas. HE, hematoxylin and eosin.

## Immunohistochemical analysis

Furthermore, we used fluorescent immunohistochemistry to determine changes in protein expression of IL-6 (Fig 6A), TNFα (Fig 6B), PANX1 (Fig 7A), and TRPA1 (Fig 7B) in rat incisor pulp tissue. Consistent with *in vitro* results, bleaching significantly increased the expression

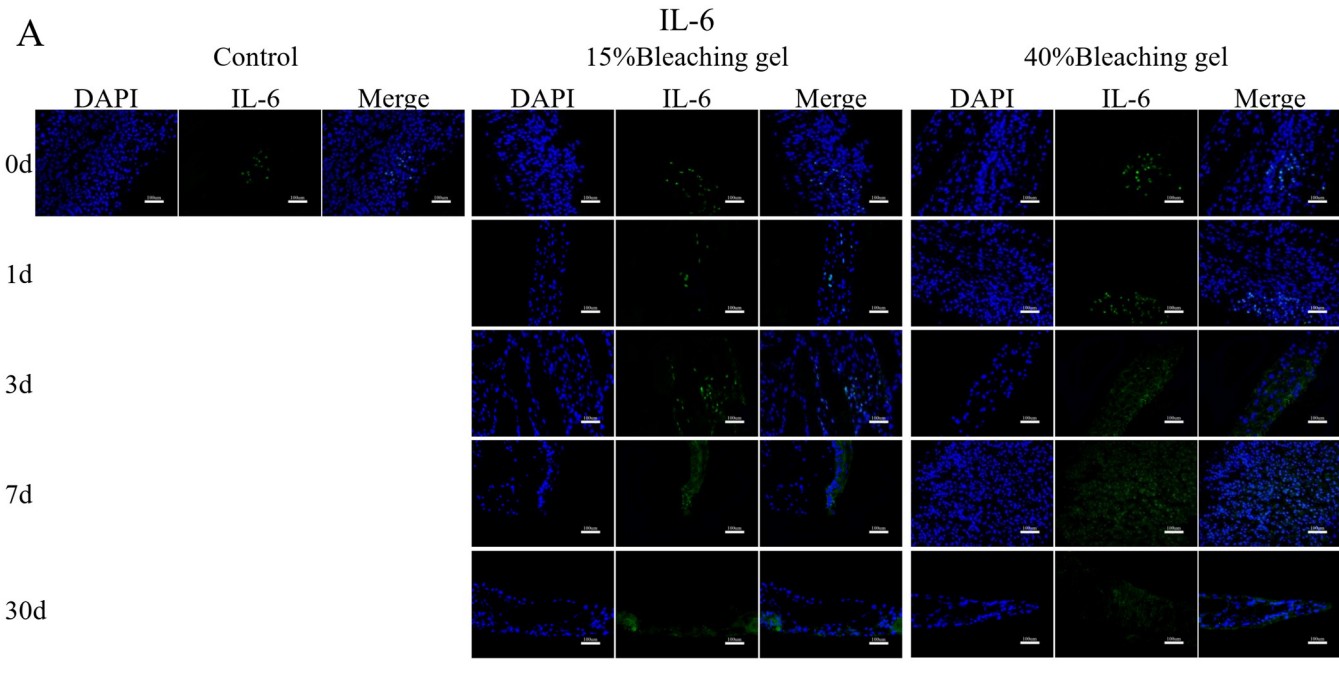

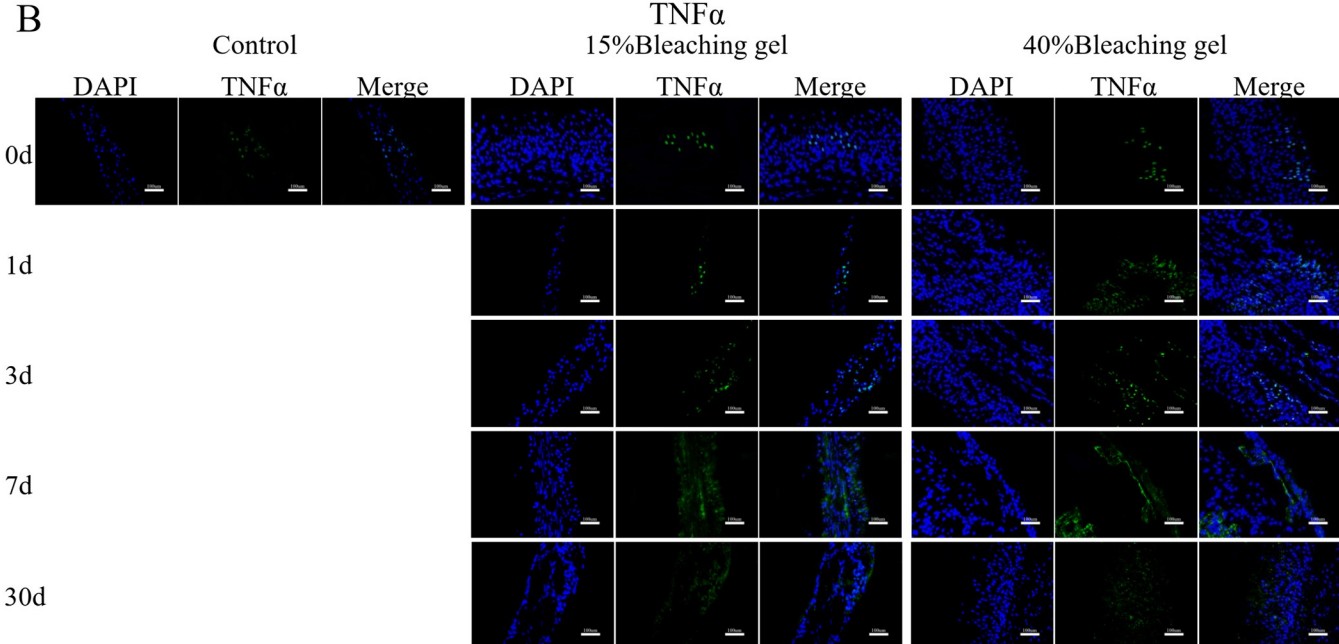

**Fig 6. Immunofluorescence staining of IL-6 and TNFα in the rat incisor after gel bleaching.** The rat incisors were bleached using the 15% or 40% gel for 90 and 30 min, respectively. After 0, 1, 3, 7, and 30 days, immunofluorescence staining was used to detect IL-6 and TNFα levels in the incisor pulp. The nuclei shown in blue were stained using DAPI. Bar = 100 μm. IL-6, interleukin 6; TNFα, tumor necrosis factor-alpha.

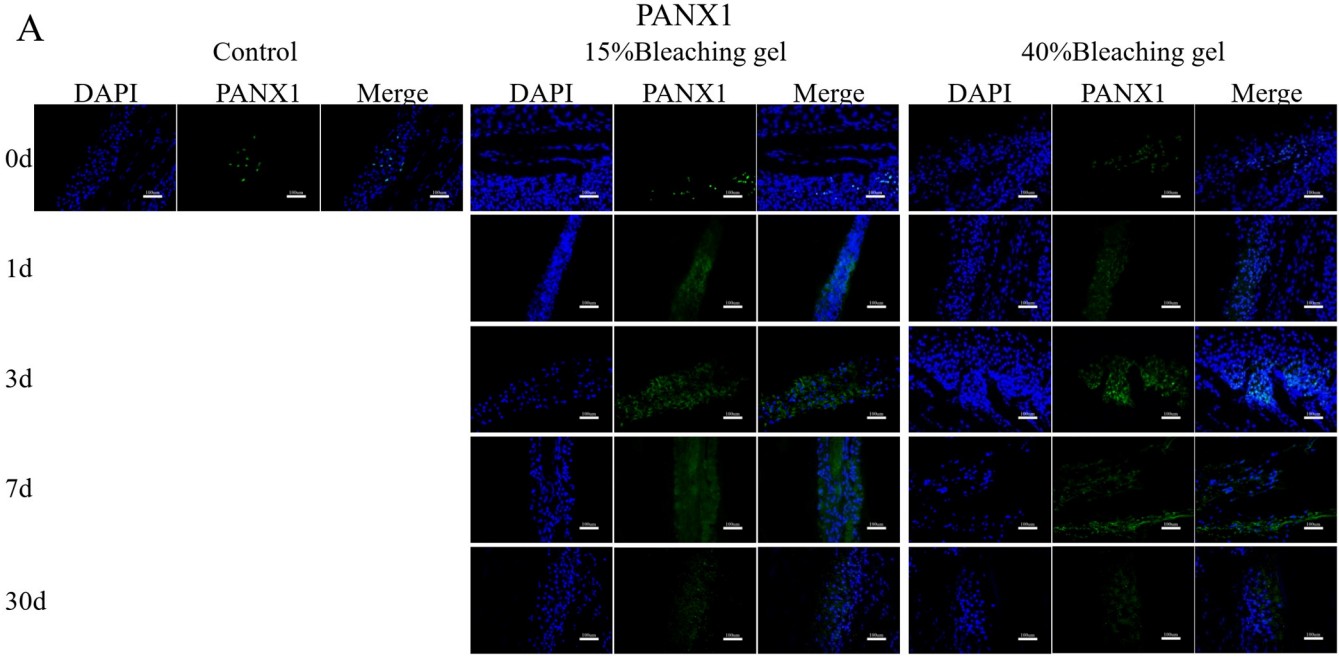

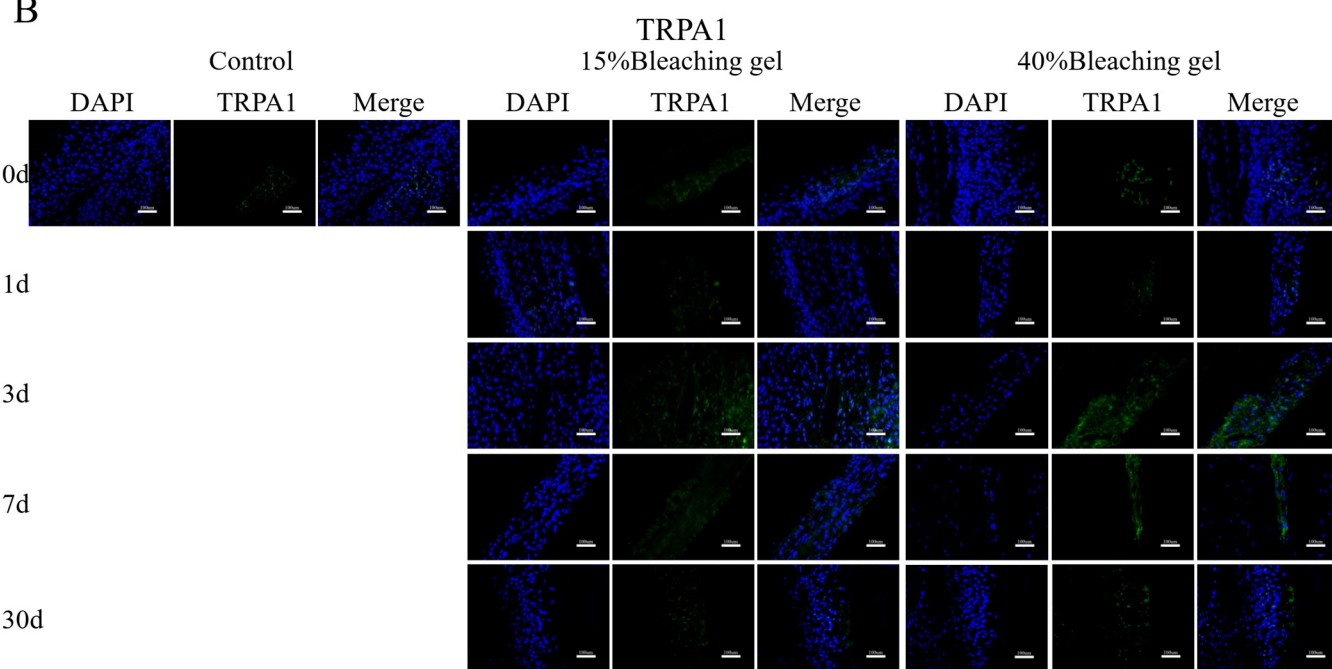

**Fig 7. Immunofluorescence staining of PANX1 and TRPA1 in rat incisors after gel bleaching.** Rat incisors were bleached using 15% or 40% gel for 90 or 30 min, respectively. After 0, 1, 3, 7, and 30 days, immunofluorescence staining was used to detected PANX1 and TRPA1 in the incisor pulp. The nuclei shown in blue were stained by DAPI. Bar = 100 μm. PANX1, Pannexin1; TRPA1, transient receptor potential ankyrin 1.

of the four proteins, with 40% gel showing more intense fluorescence. The increased protein expression was persistent until day 7 and decayed at day 30.

## Discussion

Many patients undergoing tooth-bleaching therapy present with BS, ranging from moderately severe to intolerable; therefore, there has been increasing research on the mechanisms underlying BS. This is the first study on the possible role of ROS in BS. We constructed an artificial pulp chamber as previously reported [29], with an intact human enamel-dentin block with cells attached to the dentin surface (Fig 1A). Commonly used clinical bleaching methods include in-office (40% $H_2O_2$ for 30 min) and home bleaching (15% $H_2O_2$ for 90 min), which were compared in this study.

As a progenitor population of cells that can differentiate into most cell lineages of the dental pulp tissues (odontoblast-like cells, osteoblasts-like cells, chondrocytes, skeletal muscle cells, and adipocytes) [30], DPSCs are very representative to be the ideal in vitro cell model for cytotoxic studies [31]. Its potential differentiation ability plays an important role in the process of inflammation, repair, cell necrosis, and regeneration of dental pulp tissue [32]. Consistently with previous reports [33, 34], DPSCs exhibited multi-directional differentiation ability (adipogenesis and osteogenesis) and have strong expression of stem cell surface markers (CD73, CD90, and CD105). Moreover, cultured cells were also measured by other hematopoietic markers (CD14, CD20, CD34, and CD45) to rule out the possibility that it contains hematopoietic stem cells.

There is accumulating evidence suggesting that medium ROS levels are crucial in physiological cellular processes; further, high ROS levels induce oxidative stress [35]. Increased ROS levels could be involved in inflammation and pain [36, 37]. $H_2O_2$, the most stable and common ROS form, is widely used to establish cell injury, which is directly related to the ROS concentration in the bleaching gel and the application time to enamel [38]. Using a fluorescence assay with H₂DCFDA and CellROX, we observed high ROS levels immediately after bleaching (Fig 4A and 4C), indicating that the cultured cells in contact with ROS were under oxidative stress. Furthermore, the oxidative stress intensity was directly related to the $H_2O_2$ concentration in contact with cells.

In the physiological state, there is a balance between ROS production and neutralization by antioxidant systems [27]. Disturbance of this balance results in oxidative stress. In severe cases, cell components undergo oxidative damage, which eventually compromises cell viability [39]. Numerous studies have demonstrated that DPSCs are highly sensitive to damage by $H_2O_2$ [23, 27]. We found that compared with 15% $H_2O_2$ gel and the control group, 40% $H_2O_2$ gel significantly decreased DPSC viability. NAC reduced this cytotoxicity, which further demonstrates that the gel caused cell death via ROS.

There is an association between dental pain and inflammation [40]. High-concentration bleaching gels (35–38%) smeared on rat or human teeth have been shown to cause inflammation associated with local tissue necrosis in the pulp [41, 42]. Haug et al. reported that acute dental pain caused by pulpal inflammation was positively associated with IL levels; moreover, IL-6 levels and inflammation in the pulp region could react in regions remote from the disease site [43]. Hall et al. showed that TNFα promoted inflammation and caused pain hypersensitivity in nociceptors, leading to inflammation pain similar to that in pulpitis alone [44]. Our findings indicate that bleaching gel can boost mRNA and protein expression of TNFα and IL6 in vitro and in vivo. Notably, expression of these molecules was positively correlated with intracellular ROS levels and was decreased by NAC, which is indicative of the key role of ROS in dental pain caused by the bleaching gel.

Previous studies have shown that ROS evoke pain and/or defensive reflexes in a largely TRPA1-dependent manner [15, 45]. TRPA1 acts as a nonselective cation channel induced by endogenous ROS; moreover, studies have demonstrated that TRPA1 contributes to hyperalgesia in inflammatory pain models activated by oxidizing agents [9, 10, 46]. Consistent with a previous report, we observed TRPA1 expression in DPSCs and dental pulp. To explore the molecular mechanisms underlying bleaching gel-induced pain, we conducted further experiments. Consequently, we observed a post-bleaching increase in Ca$^{2+}$, intracellular ATP, and PANX1 expression, which was weakened by NAC and was consistent with our hypothesis. Notably, there were differences between extracellular and intracellular ATP levels, as well as PANX1 expression. We suggest that excessive extracellular H$_2$O$_2$ levels in the medium of the in vitro model (Fig 1A) could cause ATP hydrolysis, and therefore reduces its extracellular levels, which is consistent with previous findings [47]. Taken together, we confirmed that bleaching gels can cause pain transmission of DPSCs through intracellular ROS. Additionally, using a rat model, we observed similar *in vivo* and *in vitro* protein expression of IL-6, TNFα, PANX1, and TRPA1. We also observed necrotic-like areas in the 40% gel bleaching group.

Bleaching gels cause dental pulp inflammation; moreover, high-concentration bleaching gels stimulate the dental pulp to a greater extent, increasing the probability of developing teeth sensitivity. Therefore, we chose a slightly lower bleach concentration in the clinical treatment.

## Conclusion

Based on our findings, it can be concluded that tooth bleaching products can produce severe oxidative stress on pulp cells, which results in intense DPSC cytotoxicity. Notably, high oxidative stress greatly induced pain conduction, including up-regulated TRPA1 expression and mediating Ca$^{2+}$ influx and ATP release into the extracellular space. This new finding contributes toward the elucidation of mechanisms underlying BS, where the strong oxidative stress on DPSCs caused by bleaching products may subsequently up-regulate TRPA1 cation channels and stimulate adjacent sensory pulp nerve fibers. This may provide a potential therapeutic target for alleviation of tooth bleaching nociception.

## Supporting information

**S1 Fig. Uncropped western original image.** (A) PANX1. (B) TNFα. (C) TRPA1. (D) β-actin. (E) IL-6.
(TIF)

**S1 Checklist. The ARRIVE guidelines 2.0: Author checklist.**
(PDF)

## Author Contributions

**Conceptualization:** Xiansheng Huang, Rong Li.

**Data curation:** Chang Chen, Xiansheng Huang, Rong Li.

**Formal analysis:** Chang Chen, Xiansheng Huang, Rong Li.

**Funding acquisition:** Xiansheng Huang, Rong Li.

**Investigation:** Chen Ding.

**Methodology:** Chen Ding, Piaopiao Huang.

**Resources:** Wenqiang Zhu.

**Software:** Wenqiang Zhu, Piaopiao Huang.

**Visualization:** Chang Chen.

**Writing – original draft:** Chang Chen.

**Writing – review & editing:** Chang Chen, Rong Li.

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
