## [Decision Letter · Decision Letter 0]

13 Jul 2021

PONE-D-21-18951

H2O2 gel bleaching induces cytotoxicity and pain conduction in dental pulp stem cells via intracellular reactive oxygen species on enamel/dentin disc

PLOS ONE

Dear Dr. li,

Thank you for submitting your manuscript to PLOS ONE. After careful consideration, we feel that it has merit but does not fully meet PLOS ONE’s publication criteria as it currently stands. Therefore, we invite you to submit a revised version of the manuscript that addresses the points raised during the review process.

Although of some interest the paper needs to be amended in some parts and requires additional experiments as specified by myself and by a referee (see for specific requirements).

We look forward to receiving your revised manuscript.

Kind regards,

Gianpaolo Papaccio, M.D., Ph.D.

Academic Editor

PLOS ONE

Additional Editor Comments:

Authors have evaluated the effects of blenching gels on DPSCs both in vitro and in vivo, demonstrating that increased ROS production in dental pulp cells lead to injuries.

The paper has some interest and although the experiments are well conducted several amendments are needed.

Introduction section is excessive and requires to be shortened.

More experiments are needed in order to evaluate ROS production in DPSCs as the Cell-Rox assays

In vivo evaluation of dental pulp after blenching treatments must be done at longer times also at longer times.

As in other papers (see BioMed Research International, 2021; Comprehensive Biomaterials, 2011) the Authors must emphasise the role of DPSCs.

Journal Requirements:

2. As part of your revisions, please update your Methods section to address the following: (1) the number of animals in each group and how you determined the sample size; (2) the sex and strain of the rats; (3) all anesthetics and analgesics administered to animals during your study (name of drug, dosage, frequency and route of administration); (4) details about monitoring parameters & humane endpoints for any animals who became severely ill during the study; (5) the rate of mortality during the study and the cause of death (if applicable); (6) lastly, please complete and submit the ARRIVE Guidelines checklist (Essential 10 version): https://arriveguidelines.org/resources/author-checklists.

Reviewers' comments:

Reviewer's Responses to Questions

**Comments to the Author**

1. Is the manuscript technically sound, and do the data support the conclusions?

Reviewer #1: Yes

2. Has the statistical analysis been performed appropriately and rigorously? 

Reviewer #1: Yes

3. Have the authors made all data underlying the findings in their manuscript fully available?

Reviewer #1: Yes

4. Is the manuscript presented in an intelligible fashion and written in standard English?

Reviewer #1: Yes

5. Review Comments to the Author

Reviewer #1: In this paper Authors evaluated the effect of blenching gels on DPSCs both in vitro and in vivo and demonstrated that the main complications are due to increased ROS production in dental pulp cells.

The paper is interesting and the experiments are well conducted.

Some changes are needed.

Introduction section should be reduced.

Authors should add other experiments to evaluate ROS production in DPSCs (Cell-Rox assay).

Evaluation of in vivo dental pulp after blenching treatments should be made also at longer times (up to 30 days).

In discussion section Authors should emphasize the role of DPSCs (BioMed Research International, 2021; Comprehensive Biomaterials, 2011)

6. PLOS authors have the option to publish the peer review history of their article (what does this mean?). If published, this will include your full peer review and any attached files.

Reviewer #1: No

---

## [Author Response · Author response to Decision Letter 0]

15 Aug 2021

For journal requirements:

1. We have modified the our manuscript according to PLOS ONE's style requirements. 

2. We have added information on animal experiments as required and submitted the ARRIVE Guidelines checklist. 

3. We apologize for using the western image that was acquired after cutting the membrane. In order to provide the entire original image without trimming, we repeated the western experiment and updated the blots of Figures 3 and 4. The original picture is shown in the additional material Figure s1.

4. We have linked ORCID iD in the revision.

Responses to the reviewers’ comments:

1. Introduction section should be reduced.

Response: Thank you for your careful review. In order to shorten the length of the introduction section, we have reduced the word count from 632 to 517.

2. Authors should add other experiments to evaluate ROS production in DPSCs (Cell-Rox assay).

Response: Thank you for your comment. We have added the Cell-Rox assay (Fig 4C and D). (line 165)

3. Evaluation of in vivo dental pulp after blenching treatments should be made also at longer times (up to 30 days).

Response: Thank you very much for your suggestion. We have added 30 days result in the revised manuscript (Figures 5, 6, and 7).

4. In discussion section Authors should emphasize the role of DPSCs (BioMed Research International, 2021; Comprehensive Biomaterials, 2011)

Response: Thank you for your suggestion. We have added the role of DPSCs in the second paragraph of the discussion section. (lines 361-371)

---

## [Editor Report · Decision Letter 1]

26 Aug 2021

H2O2 gel bleaching induces cytotoxicity and pain conduction in dental pulp stem cells via intracellular reactive oxygen species on enamel/dentin disc

PONE-D-21-18951R1

Dear Dr. li,

We’re pleased to inform you that your manuscript has been judged scientifically suitable for publication and will be formally accepted for publication once it meets all outstanding technical requirements.

Kind regards,

Gianpaolo Papaccio, M.D., Ph.D.

Academic Editor

PLOS ONE

Additional Editor Comments (optional):

The Authors have addressed the previous comments.
---

## [Editor Report · Acceptance letter]

1 Sep 2021

PONE-D-21-18951R1 

H_2_O_2_ gel bleaching induces cytotoxicity and pain conduction in dental pulp stem cells via intracellular reactive oxygen species on enamel/dentin disc 

Dear Dr. Li:

I'm pleased to inform you that your manuscript has been deemed suitable for publication in PLOS ONE. Congratulations! Your manuscript is now with our production department. 

Kind regards, 

on behalf of

Prof. Gianpaolo Papaccio 

Academic Editor

PLOS ONE